# Exploring Temporally Dynamic Data Augmentation for Video Recognition

**Taeoh Kim**[1]**, Jinhyung Kim**[2]**, Minho Shim**[1]**, Sangdoo Yun**[3]**, Myunggu Kang**[1]**,**
**Dongyoon Wee**[1]**, Sangyoun Lee**[4]

[1]NAVER Cloud, AI Tech. [2]LG AI Research [3]NAVER AI Lab [4]Yonsei University

## Abstract

Data augmentation has recently emerged as an essential component of modern training recipes for visual recognition tasks. However, data augmentation for video recognition has been rarely explored despite its effectiveness. Few existing augmentation recipes for video recognition naively extend the image augmentation methods by applying the same operations to the whole video frames. Our main idea is that the magnitude of augmentation operations for each frame needs to be changed over time to capture the real-world video's temporal variations. These variations should be generated as diverse as possible using fewer additional hyper-parameters during training. Through this motivation, we propose a simple yet effective video data augmentation framework, DynaAugment. The magnitude of augmentation operations on each frame is changed by an effective mechanism, Fourier Sampling that parameterizes diverse, smooth, and realistic temporal variations. DynaAugment also includes an extended search space suitable for video for automatic data augmentation methods. DynaAugment experimentally demonstrates that there are additional performance rooms to be improved from static augmentations on diverse video models. Specifically, we show the effectiveness of DynaAugment on various video datasets and tasks: large-scale video recognition (Kinetics-400 and Something-Something-v2), small-scale video recognition (UCF-101 and HMDB-51), fine-grained video recognition (Diving-48 and FineGym), video action segmentation on Breakfast, video action localization on THUMOS'14, and video object detection on MOT17Det.

## 1 Introduction

Data augmentation is a crucial component of machine learning tasks as it prevents overfitting caused by a lack of training data and improves task performance without additional inference costs. Many data augmentation methods have been proposed across a broad range of research fields, including image recognition (Cubuk et al., 2019; 2020; Hendrycks et al., 2019; DeVries & Taylor, 2017; Zhang et al., 2018; Yun et al., 2019; LingChen et al., 2020; Müller & Hutter, 2021), image processing (Yoo et al., 2020; Yu et al., 2020), language processing (Sennrich et al., 2016; Wei & Zou, 2019; Wang et al., 2018; Chen et al., 2020), and speech recognition (Park et al., 2019; Meng et al., 2021). In image recognition, each augmentation algorithm has become an essential component of the modern training recipe through various combinations (Touvron et al., 2021; Bello et al., 2021; Wightman et al., 2021; Liu et al., 2022). However, data augmentation for video recognition tasks has not been extensively studied yet beyond the direct adaptation of the image data augmentations.

An effective data augmentation method is required to cover the comprehensive characteristics of data. Videos are characterized by diverse, dynamic temporal variations, such as camera/object movement or photometric (color or brightness) changes. As shown in Fig. 1, dynamic temporal changes make it difficult to find out the video's category for video recognition. Therefore, it is important to make the video models effectively cope with all possible temporal variations, and it would improve the model's generalization performance.

Recent achievements in video recognition research show notable performance improvements by extending image-based augmentation methods to video (Yun et al., 2020; Qian et al., 2021). Following image transformers (Touvron et al., 2021; Liu et al., 2021a), video transformers (Fan et al., 2021;

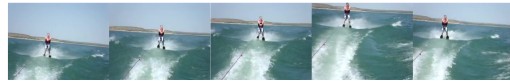 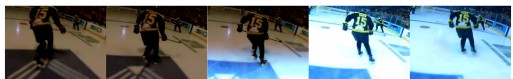

Figure 1: Applying static data augmentations has limitations in modeling temporally dynamic variations of real-world video. Figure shows examples of temporal variations in videos. (Left) *geometric changes (water skiing)* and (Right) *photometric changes (ice hockey)*. These variations are changed in an arbitrary but smooth manner.

Arnab et al., 2021; Liu et al., 2021b; Li et al., 2022) have also been applied mixture of augmentations, but they are still static over video frames. Their augmentations do not consider aforementioned temporal dynamics, simply applying the same augmentation over every frame for each video. Even though applying well-studied image augmentation (*e.g.* RandAugment (Cubuk et al., 2020)) statically on videos shows good results, there is still room for performance improvement via considering temporally dynamic factors. From this motivation, an augmentation operation on each frame should be changed dynamically rather than being static over time. To this end, we propose a simple yet effective data augmentation framework for video recognition called DynaAugment.

However, the additional temporal axis introduces new dimensions for controlling augmentation operations for videos. These increasing possibilities can cause extensive searching processes and computational costs to find the optimal augmentation policy. A unified function is required to reduce the range of the parameterization as temporal variations (either geometrically or photometrically) in the real-world videos are generally linear, polynomial, or periodic (*e.g.* moving objects, camera panning/tilting, or hand-held camera). The key factor is changing the magnitude of augmentation operation as a function of time with a simple parameterization. Based on Fourier analysis, an arbitrary signal can be decomposed into multiple basis functions. All variations, as mentioned above, can be represented by the random weighted sum of diverse-frequency sinusoidal basis functions. From this motivation, we propose a generalized sampling function called Fourier Sampling that generates temporally diverse and smooth variations as functions of time.

To verify the effectiveness of the proposed method, we conduct extensive experiments on the video recognition task, where DynaAugment reaches a better performance than the static versions of state-of-the-art image augmentation algorithms. The experimental results also demonstrate the generalization ability of DynaAugment. Specifically, recognition performances are improved in both different types of models and different types of datasets including: large-scale dataset (Kinetics-400 (Carreira & Zisserman, 2017) and Something-Something-v2 (Goyal et al., 2017)), small-scale dataset (UCF-101 (Soomro et al., 2012) and HMDB-51 (Kuehne et al., 2011)), and fine-grained dataset (Diving-48 (Li et al., 2018) and FineGym (Shao et al., 2020)). Furthermore, DynaAugment shows better transfer learning performance on the video action segmentation, localization, and object detection tasks. We also evaluate our method on the corrupted videos as an out-of-distribution generalization, especially in the low-quality videos generated with a high video compression rate. Since training with DynaAugment learns the invariance of diverse temporal variations, it also outperforms other methods in corrupted videos.

## 2 RELATED WORK

**Video Recognition**    For video recognition, 3D convolutional networks (CNNs) (Ji et al., 2012; Tran et al., 2015; Carreira & Zisserman, 2017; Hara et al., 2018; Xie et al., 2018; Tran et al., 2018; 2019; Feichtenhofer, 2020; Feichtenhofer et al., 2019) have been dominant structures that model spatio-temporal relations for videos. In another branch, Lin et al. (2019); Wang et al. (2021a) have designed temporal modeling modules on top of the 2D CNNs for efficient training and inference. Recently, transformers have proven to be strong architectural designs for video recognition with (Neimark et al., 2021; Bertasius et al., 2021; Arnab et al., 2021) or without (Fan et al., 2021) image transformer (Dosovitskiy et al., 2021) pre-training. Video transformers with attention in the local window (Liu et al., 2021b) and with convolutions (Li et al., 2022) also have shown remarkable results.

**Data Augmentation**    First, network-level augmentations (sometimes also called regularization) randomly remove (Srivastava et al., 2014; Ghiasi et al., 2018; Huang et al., 2016; Larsson et al.,

2017) or perturb (Gastaldi, 2017; Yamada et al., 2019; Verma et al., 2019; Li et al., 2020; Wang et al., 2021b; Kim et al., 2020b) in-network features or parameters. For video recognition, 3D random mean scaling (RMS) (Kim et al., 2020b) randomly changes the low-pass components of spatio-temporal features.

In image-level augmentations, after a simple data augmentation, such as a rotation, flip, crop (Krizhevsky et al., 2012), or scale jittering (Simonyan & Zisserman, 2015), algorithms that randomly remove spatial regions (DeVries & Taylor, 2017; Singh & Lee, 2017; Zhong et al., 2020) to prevent overfitting to the most discriminative parts have been proposed. To further improve the performance, a learning-based data augmentation technique called AutoAugment (Cubuk et al., 2019) is proposed and has demonstrated remarkable performances. However, AutoAugment requires many GPU hours to find the optimal augmentation policy. RandAugment (Cubuk et al., 2020) suggests that the reduction of the search space. It has produced results comparable with those of AutoAugment with only two searching parameters. Recently, search-free augmentation strategies including UniformAugment (LingChen et al., 2020) and TrivialAugment (Müller & Hutter, 2021) have shown similar performances using very simple parameterizations. They also extend the search space for augmentation operations that are inherited from those of AutoAugment to improve the performances. MixUp (Zhang et al., 2018) and CutMix (Yun et al., 2019) randomly mix the training samples; this results in both the strengths of data augmentation and label-smoothing (Szegedy et al., 2016). Extended versions (Kim et al., 2020a; Uddin et al., 2021; Kim et al., 2021) of CutMix have been designed for realistic mixing using saliency or a proper combination between samples. They have demonstrated that more realistic augmentations have superior performance, which is related to our motivation to generate realistic and diverse data augmentations for videos. VideoMix (Yun et al., 2020) is an extension of CutMix for video recognition, but the spatial version of VideoMix has shown the best performance. Furthermore, temporal versions of VideoMix are not temporally smooth. Dwibedi et al. (2020) has tried temporally varying augmentation that is closely related to our method. However, their target scopes are limited to the specific task, and they lacked a comprehensive analysis of augmentation parameters.

Because the aforementioned advanced data augmentation methods have become essential components of the modern training recipes, including CNNs (Bello et al., 2021; Wightman et al., 2021; Liu et al., 2022) and Transformers (Touvron et al., 2021), the development of strong data augmentations for video recognition is highly in demand as a fundamental building unit.

## 3 DYNAAUGMENT

### 3.1 PROBLEM DEFINITION

Given a video data $\mathbf{V} = [I_1, I_2, ..., I_T]$ that contains a total of $T$ frames, applying a single augmentation operation ($\mathtt{op}$) at frame $t$ using an augmentation strength $M$ is $\hat{I}_t = \mathtt{op}(I_t, M)$, that describes a static augmentation whose magnitude is identical over whole video frames. In DynaAugment, we define how to determine the magnitude array $\mathbf{M} = [M_1, M_2, ..., M_T]$ as a function of time $t$ to apply $\hat{I}_t = \mathtt{op}(I_t, M_t)$ as a main problem beyond the naive extension of static augmentations.

### 3.2 DYNAAUGMENT

In DynaAugment, our motivation to design the dynamic variations is that if a distribution of augmented data is similar to real videos, the generalization performance can be increased. To implement the realistic augmented videos, we set two constraints in changing the magnitudes. These constraints and their visualizations are described in Fig. 2. First, an augmentation parameter should be changed as *smoothly* as possible because arbitrary (random) changes can break the temporal consistency. Second, a set of possible arrays $\mathbf{M}$ generated during training should be as *diverse* as possible to increase the representation and generalization space of augmented samples. The static version (scalar $M$) cannot satisfy both constraints (2nd Row in Fig. 2). Meanwhile, the random variant can generate diverse variations but fails to maintain temporal consistency (3rd Row in Fig. 2). In contrast, DynaAugment (4th Row in Fig. 2) is designed for general dynamic data augmentation methodology for videos, as smooth and diverse as possible. From the perspective of the measurements to quantitatively evaluate augmentation methods proposed in Gontijo-Lopes et al. (2020), *smoothness* and *diversity* of ours

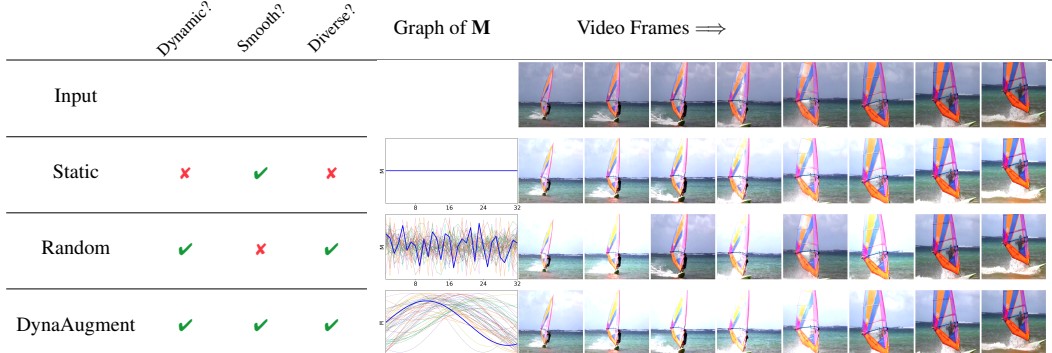

Figure 2: Comparison between static and dynamic augmentations. $X$**-axis in Graph:** temporal index from the 1st frame to the 32nd frame. $Y$**-axis in Graph:** magnitude of operations, in this example, `Brightness` operation is applied. **Blue Line in Graph:** selected $\mathbf{M}$. **Other Lines in Graph:** randomly sampled $\mathbf{M}$s in 50 trials. **1st Row:** Frames before augmentation. **2nd Row:** Static Augmentation (same brightness). **3rd Row:** Random (Per-frame) augmentations, it loses temporal consistency. **4th Row:** *DynaAugment* generates dynamic, smooth, and diverse temporal variations (getting *brighter* then *darker* again). Only 8 frames (frame rate is 4) are shown in the figure.

can be interpreted as the affinity and the diversity of Gontijo-Lopes et al. (2020), respectively. See Sec. 4.5 for the analysis.

To generate a smooth and diverse temporal array $\mathbf{M}$, we propose a novel sampling function called Fourier Sampling, as explained in the next section. In summary, DynaAugment is an extension of any image augmentation operations (*e.g.* RandAugment (Cubuk et al., 2020)) with the Fourier Sampling to generate generalized temporal arrays ($\mathbf{M}$s).

## 3.3 FOURIER SAMPLING

As mentioned in the introduction, inspired by the Fourier analysis, we make an array $\mathbf{M}$ as a weighted sum of $C$ sinusoidal basis functions to generate arbitrary smooth and partially periodic arrays.

$$\mathbf{M} = \sum_b^C w_b(\texttt{norm}(\sin[2f_b\pi\mathbf{K}[o_b : o_b + T]/(T-1)]))  \tag{1}$$

where $\mathbf{K} = [1, \ldots, 2T]$, $T$ is the number of frames, and $[:]$ is a slicing function. For each basis $b$, $[w_1, \ldots, w_c]$ are weights sampled from `Dirichlet(1.0)`, $f_b$ is a frequency that is randomly sampled from `Uniform(0.2, 1.5)`, and `norm` is a normalization operation that is determined by an amplitude $A$, which is sampled from `Uniform(0, 1)`. From $A$, a signal is min-max normalized within $[M - M * (A/1.0), M + M * (A/1.0)]$, where $M$ is a static magnitude. $o_b$ is an offset that shifts the array in the $x$-axis, which is sampled from $[0, \ldots, T-1]$. There are many possible combinations of $C$, $f_b$, $A$, and $o_b$. Through Eq. 1, we can obtain a combined array $\mathbf{M}$ as an arbitrary and smooth temporal variations. Fig. 3 shows an example of this process. A combination of diverse basis signals (Fig. 3 (a)∼(d)) produces a combined signal (Fig. 3 (e)). If more than two $\mathbf{M}$s are required, each $\mathbf{M}$ is sampled independently. More details behind this setting are in Appendix A.4 and B.5.

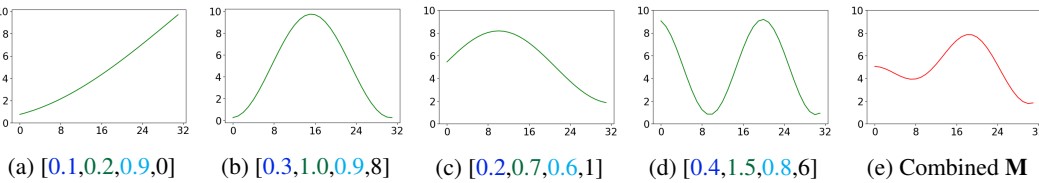

(a) [0.1,0.2,0.9,0]   (b) [0.3,1.0,0.9,8]   (c) [0.2,0.7,0.6,1]   (d) [0.4,1.5,0.8,6]   (e) Combined $\mathbf{M}$

Figure 3: An example of Fourier Sampling. $[w_b, f_b, A, o_b]$ of each basis is described.

## 4 EXPERIMENTS

**Baselines from static data augmentations** We set automatic augmentation methods used in image data as our baseline for DynaAugment. This line of work shares the following settings: (1) a pool of basic augmentation operations (*e.g.* `PIL` operations used in Cubuk et al. (2019)), (2) during training, the data-loader selects $N$ operations from the pool with a magnitude $M$ and a probability $p$ to apply that operation, and (3) algorithm-specific way to find the optimal settings of (1) and (2).

AutoAugment (AA) (Cubuk et al., 2019) finds optimal sub-policies via reinforcement learning and each sub-policy contains $N = 2$ consecutive operations with $p$ and $M$ for each operation. Since AA requires tremendous computations even in the image data, searching from scratch for video is a more challenging task to execute. We exclude these searching-based methods including Cubuk et al. (2019); Zhang et al. (2020); Lim et al. (2019) for implementation simplicity. Instead, we use more simple methods to implement such as RandAugment (RA) (Cubuk et al., 2020), TrivialAugment (TA) (Müller & Hutter, 2021), and UniformAugment (UA) (LingChen et al., 2020) to check the effects of DynaAugment as proof-of-concept. Specifically, RA uses a grid search on $N$ and $M$ per dataset and model. In this work, we use searched parameters ($N = 2$, $M = 9$) from ImageNet as a baseline. See Appendix B.2 for more searched parameters for RA. TA begins from $N = 1$ and $p = 1.0$, then randomly sample an operations as well as its magnitude $M$. Similarly, UA proposes tuning-free settings; instead, it randomly samples operation, $p$ and $M$ from the uniform distribution.

As discussed in (LingChen et al., 2020; Müller & Hutter, 2021), the search space for augmentation operations used in Cubuk et al. (2019) may not be optimal. They empirically verify that the wider (stronger) search space improves performance. Similar insight for video data is naturally required. Therefore, we extend search space with three additional operations: dynamic scale, dynamic color, and dynamic random erase (Zhong et al., 2020), which are only realized in video data. For fair comparison, we also use the extended search spaces for RA, TA, and UA. In this case, the additional operations results in the following operations: random resized crop, color jittering, and static random erasing, respectively. More details and ablation studies are described in Appendix A.3.

**Implementation Details** We reproduce the models that are trained with diverse data augmentation methods: RA (Cubuk et al., 2020), UA (LingChen et al., 2020), TA (Müller & Hutter, 2021), and our DynaAugment (DA). *Note that all other methods except DA are static versions.* For RA, TA, and UA, they are identical to DA with zero amplitude. In Fourier Sampling, we set $C = 3$ as default (See Table A8). We experiment on diverse types of video models including: 2D-CNN (Lin et al., 2019; Wang et al., 2021a), 3D-CNN (Feichtenhofer et al., 2019), 2+1D-CNN (Xie et al., 2018), efficient CNN (Feichtenhofer, 2020), transformers (Fan et al., 2021; Liu et al., 2021b), and transformers with convolutions (Li et al., 2022). We compare all methods in: 1) large-scale dataset: Kinetics-400 (Carreira & Zisserman, 2017) and Something-Something-v2 (Goyal et al., 2017), 2) small-scale dataset: UCF-101 (Soomro et al., 2012) and HMDB-51 (Kuehne et al., 2011), 3) fine-grained dataset: Diving-48 (Li et al., 2018) and FineGym (Shao et al., 2020), 4) and transfer learning tasks: video action localization on THUMOS'14 (Jiang et al., 2014), action segmentation on Breakfast (Kuehne et al., 2014), and object detection on MOT17det (Milan et al., 2016). We set all other settings and hyper-parameters following the default values of each augmentation method and model. More details are described in Appendix A.1 and A.2. Comparison with other augmentation methods are in Appendix B.2. More results for transfer learning are described in Appendix B.4.

### 4.1 RESULTS ON KINETICS-400 AND SOMETHING-SOMETHING-V2

**Datasets** Kinetics (Carreira & Zisserman, 2017) is a large-scale dataset used as a representative benchmark for many video action recognition studies. It is mainly used as a pretrain dataset for many downstream tasks. Kinetics-400 consists of 240K training and 20K validation videos in 400 action classes. Something-Something-v2 dataset (Goyal et al., 2017) contains more fine-grained temporal actions. The dataset contains 169K training and 25K validation videos with 174 action classes.

**Results on weak augmentation baselines.** Early studies of video backbone models for recognition have generally not used strong data augmentation (*e.g.* RA) or regularization (*e.g.* MixUp) as default recipes. They only contain standard data augmentations such as random temporal sampling, scale

Table 1: Results on Kinetics-400 and Something-Something-v2 dataset (Top-1 / Top-5 Accuracy). **Weak aug. baselines** contain basic augmentations: scale jittering, random crop, and random horizontal flip. **Strong aug. baselines** contain stronger augmentations: label smoothing, stochastic depth, random erasing, MixUp, CutMix, RandAugment, and repeated augmentation. (∗: ImageNet pretrained, †: trained from scratch, ⋄: Kinetics-400 pretrained)

| | Kinetics-400, *Weak aug. baselines.* | | | | | | |
|---|---|---|---|---|---|---|---|
| **Video Model** | **Baseline** | **RA** | **RA+DA** | **TA** | **TA+DA** | **UA** | **UA+DA** |
| TSM-R50-8×8∗ | 74.2 / 91.4 | 74.4 / 91.5 | **75.1 / 92.2** | 75.5 / 92.4 | **75.8 / 92.6** | 75.4 / 92.2 | **75.7 / 92.8** |
| TDN-R50-8×5∗ | 76.9 / 92.9 | 77.8 / 93.6 | **77.9 / 93.7** | 78.3 / 93.6 | **78.5 / 94.0** | 78.2 / 93.7 | **78.7 / 94.3** |
| SlowOnly-R50-8×8† | 74.8 / 91.6 | 74.7 / 91.3 | **75.4 / 92.0** | 75.6 / 92.2 | **76.1 / 92.7** | 75.7 / 92.1 | **76.2 / 92.6** |
| SlowFast-R50-8×8† | 76.8 / 92.6 | 77.3 / 93.2 | **78.0 / 93.9** | 78.5 / 93.6 | **79.1 / 94.1** | 78.3 / 93.5 | **79.0 / 94.0** |
| X3D-M-16×5† | 76.0 / 92.4 | 76.3 / 92.6 | **76.9 / 93.1** | 76.6 / 92.9 | **77.0 / 93.3** | 76.6 / 93.1 | **77.1 / 93.4** |
| Swin-Tiny-32×2∗ | 78.9 / 93.9 | 79.4 / 94.2 | **80.0 / 94.3** | 79.6 / 94.2 | **80.2 / 94.5** | 79.6 / 94.2 | **80.1 / 94.4** |
| Swin-Base-32×2∗ | 81.1 / 94.7 | 81.7 / 95.0 | **82.0 / 95.1** | 81.8 / 95.0 | **82.1 / 95.2** | 82.0 / 95.1 | **82.3 / 95.2** |

| | Kinetics-400, *Strong aug. baselines.* | | | | | | |
|---|---|---|---|---|---|---|---|
| | **Baseline** | **Baseline with DA** | **No Aug.** | **RA Only** | **RA+DA Only** | | |
| Swin-Tiny-32×2† | 77.1 / 93.1 | **77.6 / 93.3** | 73.9 / 90.5 | 76.4 / 92.6 | **76.9 / 92.9** | | |
| MViT-S-16×4† | 76.1 / 92.3 | **76.7 / 92.9** | 69.4 / 88.0 | 73.0 / 91.0 | **73.9 / 91.5** | | |
| Uniformer-S-16×8∗ | 80.1 / 94.4 | **80.9 / 94.8** | 76.6 / 92.2 | 79.5 / 93.9 | **80.3 / 94.4** | | |
| Uniformer-B-16×4∗ | 81.6 / 94.7 | **82.0 / 95.0** | - | - | - | - |

| | Something-Something-v2, *Weak aug. baselines.* | | | | | | |
|---|---|---|---|---|---|---|---|
| | **Baseline** | **RA** | **RA+DA** | **TA** | **TA+DA** | **UA** | **UA+DA** |
| TSM-R50-16∗ | 63.2 / 88.0 | 65.4 / 89.6 | **66.0 / 89.7** | 65.7 / 89.3 | **66.1 / 89.8** | 65.7 / 89.5 | **66.3 / 90.0** |
| TDN-R50-8×5∗ | 64.0 / 88.8 | 64.9 / 88.4 | **65.6 / 89.6** | 66.1 / 89.9 | **66.6 / 90.2** | 66.3 / 89.9 | **66.6 / 90.2** |
| SlowFast-R50-8×8⋄ | 61.4 / 85.8 | 63.1 / 87.3 | **65.0 / 89.0** | 63.5 / 89.0 | **64.8 / 88.9** | 63.1 / 88.7 | **64.0 / 88.9** |
| SlowFast-R50-16×8⋄ | 63.0 / 88.5 | 64.4 / 88.7 | **65.5 / 89.5** | 64.7 / 88.6 | **65.8 / 89.8** | 64.6 / 88.5 | **65.7 / 89.6** |

| | Something-Something-v2, *Strong aug. baselines.* | |
|---|---|---|
| | **Baseline** | **Baseline with DA** |
| Swin-Base-32×2⋄ | 69.9 / 92.7 | **70.7 / 93.0** |
| Uniformer-B-16×4⋄ | 70.2 / 92.5 | **71.0 / 93.2** |

jittering, random crop, and random flip. We describe experimental results on those models with augmentation methods in Table 1 with title of *weak aug. baselines*.

In Kinetics-400, using static automatic augmentation consistently improves the performances, for example, TA boosts the performance from 0.6% (X3D-M) to 1.7% (SlowFast). Using dynamic extension, the performances are further boosted up to 0.6%. Interestingly, the breadth of performance improvement tends to increase as the model's capacity increases, the number of input frames increases, or the model is not pretrained. The improvements are similar in Something-Something-v2, but the improvement gaps became larger. This demonstrates the effect of DA in more dynamic dataset.

**Results on strong augmentation baselines.** Recent models (Liu et al., 2021b; Fan et al., 2021; Li et al., 2022) use a combination of strong augmentations and regularizations. For example, label smoothing (Szegedy et al., 2016), stochastic depth (Huang et al., 2016), random erasing (Zhong et al., 2020), MixUp (Zhang et al., 2018), CutMix (Yun et al., 2019), RandAugment (Cubuk et al., 2020), and repeated augmentation (Hoffer et al., 2020) are used. From this recipe, we extend RandAugment into DynaAugment but maintain other augmentations and training configurations.

The results in Table 1 (with title of *strong aug. baselines*) show that the clear improvements by dynamic extensions in all cases from 0.4% to 0.8% Top-1 accuracy although the recipes are tuned for RA. We also conduct experiments without other augmentations but only RA is used (RA Only). From the results, RA greatly improves the performance from the *no-aug* baseline, and DA further boosts the performances up to 0.9%. Approximately 1% performance improvement can be achieved by increasing the model size (*e.g.* SlowFast-R50: 77.0% -> SlowFast-R101: 77.9% in Kinetics-400), which we can achieve without the additional inference cost when we use DA.

## 4.2 RESULTS ON OTHER DATASETS

**Datasets and Settings** We use UCF-101 (Soomro et al., 2012) and HMDB-51 (Kuehne et al., 2011) datasets for verifying the effectiveness of the data augmentations in the small-scale dataset. UCF-

Table 2: Results on UCF-101, HMDB-51, Diving-48, and Gym288 dataset (Top-1 / Top-5 Accuracy).
‡: Per-class evaluation.

| Dataset | Baseline | RA | RA+DA | TA | TA+DA | UA | UA+DA |
|---|---|---|---|---|---|---|---|
| UCF-101 | 72.0 / 90.6 | 82.8 / 96.4 | **85.3 / 97.2** | 81.6 / 96.3 | **82.8 / 95.7** | 82.5 / 96.1 | **82.9 / 95.6** |
| HMDB-51 | 40.1 / 72.6 | 48.5 / 78.4 | **54.8 / 83.7** | 47.7 / 78.9 | **51.9 / 80.7** | 49.9 / 79.8 | **52.4** / 78.9 |
| Diving-48 | 62.8 / 93.6 | 63.6 / 93.8 | **70.5 / 95.6** | 68.3 / 95.1 | **72.5 / 95.7** | 67.8 / 95.0 | **70.0 / 94.9** |
| Gym288‡ | 54.9 / 83.9 | 52.6 / 80.6 | **55.2 / 82.0** | 54.4 / 80.9 | **55.6 / 83.2** | 53.4 / 82.4 | **56.1 / 83.5** |

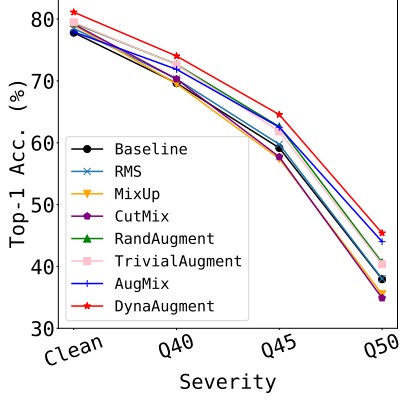

(a) Severity-Performance Plot

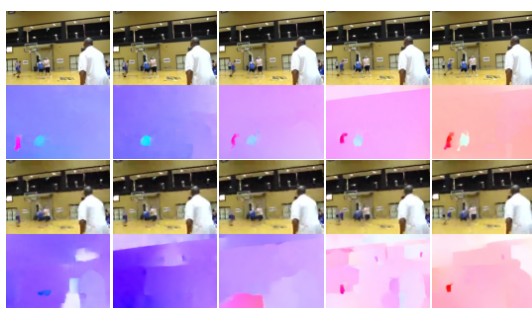

(b) Visualization

Figure 4: (a) Performance drop on spatio-temporally corrupted videos. $Q$ indicates the quantization parameter of compression, and high $Q$ indicates severe degradation. (b) Visualization of temporal variations using optical flow. Five frames above: original video, below: compressed video.

101 (Soomro et al., 2012) contains 13,320 trimmed videos in 101 action classes. HMDB-51 (Kuehne et al., 2011) contains 6,849 videos in 51 action classes. Both dataset consist of three train/test splits, and we use the first split.

Diving-48 (Li et al., 2018) and FineGym (Shao et al., 2020) datasets are used as fine-grained action recognition. Diving-48 dataset contains 18,404 videos in 48 fine-grained diving action classes. Gym288 dataset in FineGym contains 38,980 videos in 288 fine-grained gymnastic action classes. Since all videos share similar background and static contexts, the model should have strong temporal understanding to catch the fine-grained action of an athlete (Choi et al., 2019). For Gym288, because the distribution of the dataset is heavily imbalanced, we evaluate per-class accuracy. The S3D-G (Xie et al., 2018) model is used and trained from scratch to check the clear effect of data augmentation for all experiments. More results on other baselines are in Appendix B.3.

**Results** Table 2 shows the results on the four datasets. The results show that DA consistently outperforms its static counterpart. Specifically, in UCF-101 dataset, being dynamic over RA improves up to 2.5% in top-1 accuracy, and 13.3% over baseline. The results in HMDB-51 show a similar tendency. In Diving-48 dataset, DA increase the performance by 6.9% over RA that indicate DA also significantly improves fine-grained representations of the model. The results indicate that better data augmentation is critical to the performance improvements in the small-scale dataset compared to the large-scale dataset because augmented data is a kind of missing training data. In Gym288 dataset, the results are not effective as in the other datasets; static augmentation even degrades the baseline performance. Nevertheless, DA improves the accuracy up to 1.2%.

### 4.3 RESULTS ON CORRUPTED VIDEOS.

To test and compare the corruption robustness of the data augmentation algorithms in video recognition due to the difficulty of quantifying video variations, we make a validation set from the subset of Kinetics through *video compression*. In a mobile environment, the quality of videos can become extremely degraded due to the network conditions, and it can be dynamic (See Fig. 4 (b)) due to

Table 3: Transfer learning to video action localization task. Results on THUMOS'14 dataset.

| Backbone | SlowOnly-50 | | | | | | SlowFast-50 | | | | | |
|---|---|---|---|---|---|---|---|---|---|---|---|---|
| Augmentation | 0.3 | 0.4 | 0.5 | 0.6 | 0.7 | Avg mAP | 0.3 | 0.4 | 0.5 | 0.6 | 0.7 | Avg mAP |
| None | 51.1 | 44.2 | 34.2 | 24.7 | 15.3 | 33.9 | 55.7 | 49.9 | 41.8 | 32.4 | 22.8 | 40.5 |
| UA | 53.4 | 46.2 | 36.8 | 27.1 | 17.4 | 36.2 | 56.2 | 51.0 | 43.0 | 33.7 | 23.2 | 41.4 |
| UA+**DA** | **53.8** | **47.0** | **37.8** | **28.2** | **18.5** | **37.1** | **57.3** | **51.5** | **43.6** | **33.9** | **23.5** | **42.0** |

Table 4: Transfer learning to video action segmentation and video object detection tasks.

(a) Action segmentation results on Breakfast dataset.

| | | F1 | | |
|---|---|---|---|---|
| Configuration | Acc. | @0.10 | @0.25 | @0.50 |
| SlowOnly-50 | 59.0 | 54.7 | 49.2 | 37.6 |
| SlowOnly-50+UA | 62.3 | 59.1 | 53.7 | 40.8 |
| SlowOnly-50+UA+**DA** | **64.7** | **60.6** | **55.5** | **43.2** |

(b) Object detection results on MOT17det dataset.

| Configuration | AP | AP50 | AP75 | APs |
|---|---|---|---|---|
| Swin-T | $30.3_{\pm0.9}$ | 63.7 | 25.1 | 4.1 |
| Swin-T+TA | $29.1_{\pm1.3}$ | 62.1 | 23.2 | 3.7 |
| Swin-T+TA+**DA** | $\mathbf{31.3_{\pm0.6}}$ | **65.3** | **26.3** | **5.7** |

the adaptive rate control. We use H.264/AVC (Wiegand et al., 2003) to compress the videos with different levels of QP (quantization parameters). More details are in Appendix A.7.

Fig. 4 (a) presents the results as function of each corruption's severity. Automated data augmentation such as RA (Cubuk et al., 2020) and TA (Müller & Hutter, 2021) show good robustness over baseline and regularization methods including RMS (Kim et al., 2020b), MixUp (Zhang et al., 2018) and CutMix (Yun et al., 2019). AugMix (Hendrycks et al., 2019) is designed for the corruption robustness and also shows good performances in hard corruptions, but it falls behind with clean and less corrupted inputs. In contrast, DA shows best results over every severity level demonstrating good clean accuracy and corruption robustness. Because in the corrupted videos, it is observed that the spatio-temporal variations between frames are generally increased as shown in Fig. 4 (b), augmented representation of DA can be generalized into corrupted videos. (See also qualitative results in Appendix B.8.)

## 4.4 TRANSFER LEARNING

**Video Action Localization** THUMOS'14 dataset (Jiang et al., 2014) has 20 class subsets of the untrimmed videos. The videos contain long sequences (up to 26 min) with sparse action segments, and the goal is to classify video-level actions and localize the temporal actions. We experiment on fully-supervised setting on G-TAD (Xu et al., 2020). The features are extracted from Kinetics-400 pretrained backbones. The results in Table 3 show that the performance (mAP@IoU) is improved by the pretrained model using DA training compared to the model using UA training. This indicates that DA learns better representations in terms of temporal sensitivity from temporal variations.

**Video Action Segmentation** Breakfast dataset (Kuehne et al., 2014) contains 1,712 untrimmed videos with temporal annotations of 48 actions related to breakfast preparation. Unlike action localization, all frames are densely classified into pre-defined class. We experiment using MS-TCN (Farha & Gall, 2019) from the features that are extracted from Kinetics-400 pretrained SlowOnly-50. The results in Table 4a also demonstrate the superior performance of DA-pretrained features.

**Video Object Detection** For video object detection, MOT17 detection benchmark (Milan et al., 2016) is used. We follow the previous practice (Zhou et al., 2020) to split the MOT17 training set into two parts, one for training and the other for validation. As a downstream task of the video recognition, from the CenterNet (Zhou et al., 2019)-based detection framework, we substitute the backbone to our 3D video backbone. Swin-Tiny is used as our 3D backbone, and the results in Table 4b show that the detection performance (AP) is clearly increased, especially in the hard cases (AP75 and APs). With repeated experiments, baseline (Swin-Tiny) and TA show results with high variability confirmed by the high standard deviation (std) of results. We attribute the inferiority of TA to this variability, with full results available in Appendix B.4. Static augmentation may not be proper for some datasets, as observed in Gym288 result of Table 2. Nevertheless, DA not only gives consistently superior results, but the std is also reduced compared to baseline and TA.

Table 5: Ablation studies and analysis on diverse augmentation settings. For (b) and (c), RA is used, and TA is used for Diving-48 in (b).

(a) Results on different search space. Org.: original image search space, Mod.: modified video search space, Wide: wide search space used in TA.

| Dataset | Config. | Space | Top-1/Top-5 |
|---------|---------|-------|-------------|
| UCF-101 | Baseline | - | 72.0 / 92.6 |
| | RA | Org. | 79.7 / 94.3 |
| | RA+DA | Org. | **83.0 / 96.2** |
| | RA | Mod. | 82.8 / 96.4 |
| | RA+DA | Mod. | **85.3 / 97.2** |
| Kinetics-100 | Baseline | - | 66.6 / 85.5 |
| | TA | Org. | 71.1 / 88.4 |
| | TA | Wide | 72.3 / 89.4 |
| | TA+DA | Wide | **73.2 / 90.0** |
| | TA | Wide+Mod. | 72.7 / 89.9 |
| | TA+DA | Wide+Mod. | **73.7 / 90.5** |

(b) Results on different dynamic variations.

| Config. | UCF-101 | Diving-48 |
|---------|---------|-----------|
| Baseline | 72.0 / 92.6 | 62.8 / 93.6 |
| Static | 82.8 / 96.4 | 68.3 / 95.1 |
| Linear | 80.1 / 94.8 | 69.2 / **95.7** |
| Sinusoidal | 81.4 / 95.5 | 71.8 / 94.9 |
| Random | 76.9 / 91.7 | 71.0 / 94.0 |
| Ours | **85.3 / 97.2** | **72.5 / 95.7** |

(c) Affinity (Aff.) and diversity (Div.) measurement on diverse dynamic variations. Measured on UCF-101.

| Config. | Aff. | Div. | $\Delta$Top-1. |
|---------|------|------|--------|
| Static | 0.93 | 1.43 | +10.8 |
| Linear | **0.96** | 1.39 | +7.9 |
| Random | 0.63 | 1.46 | +4.9 |
| Ours | **0.96** | **1.59** | **+13.3** |

## 4.5 ABLATION STUDY AND DISCUSSION

**Search Space** As mentioned above, augmentation spaces used for image recognition may not be optimal for video due to the video's dynamic nature. Table 5 (a) shows the result of different augmentation search spaces in UCF-101 and Kinetics-100 (used in (Chen et al., 2021)) datasets. The results show that the improvements of DA are not due to the modified search space; instead, making temporally dynamic is the more critical factor.

**Smoothness** As shown in Fig. 2, smoothness and diversity are important factors in designing the natural temporal variations. In Table 5 (b), the result shows that although static augmentation boosts the performance in UCF-101 and Diving-48, dynamic variations further boost the performance. However, simple dynamic extensions (DA without Fourier Sampling) such as naive linear or sinusoidal lack diversity, and random variation lacks smoothness. These examples are visualized in Appendix A.5. In UCF-101, they underperform the static version. In contrast, in the more temporally-heavy Diving-48 dataset, making variations dynamic is more critical.

**Affinity and Diversity** To check *why* the performances of DA are remarkable, two qualitative measures for data augmentation proposed in Gontijo-Lopes et al. (2020) are used. The first one is the *affinity* which describes the distribution shift from the clean dataset, and the second one is the *diversity* which describes the number of unique training samples. See Appendix A.6 for more details. The results in Table 5 (c) show that DA has both high affinity and diversity. Meanwhile, the static version has low affinity, the naive linear version has low diversity, and the random version has very low affinity. This indicates that the dynamic variations produced by Fourier Sampling are natural.

## 5 CONCLUSION AND LIMITATION

Extensive experimental results indicate that the key factor in increasing video recognition performance is to make the augmentation process as temporally dynamic, smooth, and diverse as possible. We implement such properties as a novel framework called DynaAugment. It shows that there is room for performance to be improved on various video recognition tasks, models, and datasets over the static augmentations. We expect DynaAugment to be an important component of improved training recipes for video recognition tasks. Lack of finding these recipes combined with other augmentation and regularization methods is one of our limitation in this paper. We reserve it for future work because it requires considerable effort and resources. We also hope to apply DynaAugment beyond supervised learning (*e.g.* self-supervised learning in Appendix B.9).

## REPRODUCIBILITY STATEMENT

The code will be available at https://github.com/clovaai/dynaaug to ensure reproducibility. Any details not included in the source code are described in Appendix A.

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

# A    IMPLEMENTATION DETAILS

## A.1    DETAILED CONFIGURATIONS

Table A1: Detailed Configurations for Kinetics-400 models. SGD(M) is SGD with momentum optimizer, the numbers in step schedule are epochs for decaying. For train/test sampling, dense sampling uniformly samples in the whole temporal range, while segment sampling extracts frames from each local segment. For test view, spatial×temporal cropping is used. For TSM, learning rate is decayed at 50 and 100 epoch, and for TDN, learning rate is decayed at 50, 100, and 125 epoch. For the official and reproduced results, Top-1 accuracies (%) are described.

| Video Model | TSM | TDN | SlowOnly | SlowFast | X3D-M | Swin-T | Swin-B | Swin-T | MViT-S | Uniformer-S | Uniformer-B |
|---|---|---|---|---|---|---|---|---|---|---|---|
| Pretrain | ImageNet | ImageNet | - | - | - | ImageNet | ImageNet | - | - | ImageNet | ImageNet |
| Frame/Rate | 8×8 | 40 | 8×8 | 32×2 | 16×5 | 32×2 | 32×2 | 32×2 | 16×4 | 16×8 | 16×4 |
| Epochs | 150 | 150 | 300 | 300 | 300 | 60 | 60 | 200 | 200 | 110 | 110 |
| Batch Size | 64 | 128 | 64 | 128 | 128 | 64 | 32 | 64 | 128 | 32 | 40 |
| Learning Rate | 0.01 | 0.02 | 0.2 | 0.2 | 0.2 | 1e-03 | 1e-03 | 4e-04 | 8e-04 | 1e-04 | 1.25e-4 |
| Optimizer | SGD(M) | SGD(M) | SGD(M) | SGD(M) | SGD(M) | AdamW | AdamW | AdamW | AdamW | AdamW | AdamW |
| Schedule | Step | Step | Cosine | Cosine | Cosine | Cosine | Cosine | Cosine | Cosine | Cosine | Cosine |
| Warmup | - | - | 40 | 40 | 35 | 5 | 5 | 30 | 30 | 10 | 10 |
| Weight Decay | 1e-04 | 1e-04 | 1e-04 | 1e-04 | 5e-05 | 0.02 | 0.05 | 0.05 | 0.05 | 0.05 | 0.05 |
| DropPath | - | - | - | - | - | 0.1 | 0.3 | 0.1 | 0.1 | 0.1 | 0.3 |
| DropOut | 0.5 | 0.5 | 0.5 | 0.5 | 0.5 | 0.5 | 0.5 | 0.5 | 0.5 | - | - |
| Train Sampling | Dense | Segment | Dense | Dense | Dense | Dense | Dense | Dense | Dense | Dense | Dense |
| Test Sampling | Dense | Segment | Dense | Dense | Dense | Dense | Dense | Dense | Dense | Dense | Dense |
| Test View | 1×10 | 3×10 | 3×10 | 3×10 | 3×10 | 3×4 | 3×4 | 3×4 | 1×5 | 1×4 | 1×4 |
| Repeat Aug. | - | - | - | - | - | - | - | 2 | 2 | 2 | 2 |
| RandAug | - | - | - | - | - | - | - | (7, 4) | (7, 4) | (7, 4) | (7, 4) |
| MixUp | - | - | - | - | - | - | - | 0.8 | 0.8 | 0.8 | 0.8 |
| CutMix | - | - | - | - | - | - | - | 1.0 | 1.0 | 1.0 | 1.0 |
| Label Smoothing | - | - | - | - | - | - | - | 0.1 | 0.1 | 0.1 | 0.1 |
| Official | 74.1 | 76.6 | 74.8 | 77.0 | 76.0 | 78.8 | 80.6 | N/A | 76.0 | 80.8 | 82.0 |
| Reproduced | 74.2 | 76.9 | 74.8 | 76.8 | 76.0 | 78.9 | 81.1 | 77.1 | 76.1 | 80.1 | 81.6 |

We use PyTorch (Paszke et al., 2017) for the experiments. All our experiments are conducted in a single machine with 8 × A100 or 8 × V100 GPUs. We implement our augmentation module on top of the public codebases (TSM[1], TDN[2], SlowOnly[3], SlowFast[3], X3D[3], MViT[3], Swin[4], and Uniformer[5]) for video action recognition.

We try to follow the original training recipes except for the training epochs for all models. This choice follows the previous observations that strong augmentation requires more training iterations for convergence. In Table A1, we describe all hyper-parameters and compare our reproduced baselines with the official results in Kinetics-400 (Carreira & Zisserman, 2017) dataset. These slight differences might be caused by some modifications, such as batch size (or batch per GPU) or training epochs. However, we show the generalization effects of DynaAugment that boost all performances across the baselines.

In Table A2, we describe all hyper-parameters and compare our reproduced baselines with the official results in the other datasets, such as Something-Something-v2 (Goyal et al., 2017), UCF-101 (Soomro et al., 2012), HMDB-51 (Kuehne et al., 2011), Diving-48 (Li et al., 2018), and Gym288 (Shao et al., 2020). Note that UCF-101 and HMDB-51 shares all settings, and Diving-48 and Gym288 also shares all settings. For the small dataset, such as UCF-101, HMDB-51, Diving-48, and Gym288, we intentionally train the models from scratch to check the clear effects of the data augmentation. For improved results on these datasets (UCF/HMDB) using ImageNet or Kinetics pre-training, please refer to Section B.3.

## A.2    TRANSFER LEARNING DETAILS

For all transfer learning experiments, from the recognition models that are pre-trained with or without DynaAugment (or TrivialAugment/UniformAugment), features are pre-extracted. From the features, all localization (segmentation or detection) head parts are trained with the downstream datasets.

---

[1] https://github.com/mit-han-lab/temporal-shift-module
[2] https://github.com/MCG-NJU/TDN
[3] https://github.com/facebookresearch/SlowFast
[4] https://github.com/SwinTransformer/Video-Swin-Transformer
[5] https://github.com/Sense-X/UniFormer/tree/main/video_classification

Table A2: Detailed Configurations for Something-Something-v2 (Goyal et al., 2017) (Sth-v2), UCF-101 (Soomro et al., 2012), HMDB-51 (Kuehne et al., 2011), Diving-48 (Li et al., 2018) (DV), and Gym288 (Shao et al., 2020) (Gym) models. SGD(M) is SGD with momentum optimizer, the numbers in step schedule are epochs for decaying. For train/test sampling, dense sampling uniformly samples in the whole temporal range, while segment sampling extracts frames from each local segment. For test view, spatial×temporal cropping is used. For the official and reproduced results, Top-1 accuracies (%) are described.

| **Video Model** | TSM | TDN | SlowFast | SlowFast | Swin-B | Uniformer-B | S3D-G | S3D-G |
|---|---|---|---|---|---|---|---|---|
| Dataset | Sth-v2 | Sth-v2 | Sth-v2 | Sth-v2 | Sth-v2 | Sth-v2 | UCF/HMDB | DV/Gym |
| Pretrain | ImageNet | ImageNet | K400 | K400 | K400 | K400 | - | - |
| Frame/Rate | 16 | 40 | 32×2 | 64×2 | 32×2 | 16×4 | 32×2 | 32×2 |
| Epochs | 90 | 80 | 40 | 40 | 60 | 60 | 200 | 200 |
| Batch Size | 64 | 128 | 64 | 64 | 32 | 40 | 64 | 64 |
| Learning Rate | 0.01 | 0.02 | 0.12 | 0.12 | 3e-04 | 2.5e-04 | 0.025 | 0.025 |
| Optimizer | SGD(M) | SGD(M) | SGD(M) | SGD(M) | AdamW | AdamW | SGD(M) | SGD(M) |
| Schedule | Step [30, 60] | Step [40, 60, 70] | Step [26, 33] | Step [26, 33] | Cosine | Cosine | Cosine | Cosine |
| Warmup | - | - | 0.19 | 0.19 | 2.5 | 5 | - | - |
| Weight Decay | 1e-04 | 5e-04 | 1e-06 | 1e-06 | 0.05 | 0.05 | - | - |
| DropPath | - | - | - | - | 0.4 | 0.4 | - | - |
| DropOut | 0.5 | 0.5 | 0.5 | 0.5 | - | - | 0.8 | 0.8 |
| Train Sampling | Segment | Segment | Segment | Segment | Segment | Segment | Dense | Segment |
| Test Sampling | Segment | Segment | Segment | Segment | Segment | Segment | Dense | Segment |
| Test View | 3×2 | 3×1 | 3×1 | 3×1 | 3×1 | 3×1 | 3×10 | 3×1 |
| Repeat Aug. | - | - | - | - | - | 2 | - | - |
| RandAug | - | - | - | - | (7, 4) | (7, 4) | - | - |
| MixUp | - | - | - | - | - | 0.8 | - | - |
| CutMix | - | - | - | - | - | 1.0 | - | - |
| Label Smoothing | - | - | - | - | 0.1 | 0.1 | - | - |
| Official | 63.4 | 64.0 | N/A | 63.0 | 69.6 | 70.4 | N/A | N/A |
| Reproduced | 63.2 | 64.0 | 61.4 | 63.0 | 69.6 | 70.2 | - | - |

For all transfer learning algorithms, such as G-TAD[6] (Xu et al., 2020) and MS-TCN[7] (Farha & Gall, 2019), we follow the original implementation without configuration changes. One exception is the feature extraction stage we describe below.

For G-TAD (Xu et al., 2020), features are extracted in every frame via sliding window whose window size is 9 frames. For MS-TCN (Farha & Gall, 2019), features are extracted in every frame via sliding window whose window size is 21 frames, with 15 frame-per-second.

For video object detection, we use CenterNet (Zhou et al., 2019) as a detection head. The backbone is replace with our video backbones. The training losses are identical to Zhou et al. (2019), but the configurations are changed: batch size 8, total 30 epochs, weight decay 0.05, AdamW (Loshchilov & Hutter, 2017) optimizer, learning rate 1e-04, and step decay 0.1 at 20 epochs.

Our motivation is to verify the effects of data augmentation for the upstream model, whose downstream task has not been explored much and whose downstream dataset is quite different from the upstream dataset, such as MOT. Therefore, we verified our purpose that DynaAugment is superior to the model trained without augmentation and the static-augmented model.

## A.3   DETAILS ON THE EXTENDED SEARCH SPACE

As discussed in LingChen et al. (2020); Müller & Hutter (2021), the search space for augmentation operations used in Cubuk et al. (2019) may not be optimal. They empirically verify that the wider (stronger) search space improves performance. Similar insight for video data is naturally required. Therefore, we extend search space with three additional operations: dynamic scale, dynamic color, and dynamic random erase (Zhong et al., 2020), which are only realized in video data. They are visualized in Fig A5. For dynamic scale, we first resize an image then add padding (scale-down) or crop centers (scale-up). The magnitude range of scale change is [0.667, 1.5], and [0.5, 2.0] for wide search space. For dynamic color, we modify an image's hue using `torchvision` function. The magnitude range is [-0.1, 0.1], and [-0.3, 0.3] for wide. For dynamic random erase, we follow random erasing in image (Zhong et al., 2020), but the box size varies across frames. The magnitude range is

---

[6]`https://github.com/frostinassiky/gtad`
[7]`https://github.com/yabufarha/ms-tcn`

[10, 30], and [10, 60] for wide. The numbers in here means the ratio of the area of the box to the total image area. These three operations are related to the real-world variations, such as, geometric variations (object movements, camera movements, *e.g.*, zoom-in), photometric variations (`color` in AA (Cubuk et al., 2019)'s augmentation search space only change the saturation of an image), and time-varying occlusions.

The results in Table A3 show that adding each operation has little differences in performances, but the performances are improved slightly when adding all three operations. This is because we already have 14 operations (first used in AA) in the pool, and the probability that the added operations will be sampled is relatively small.

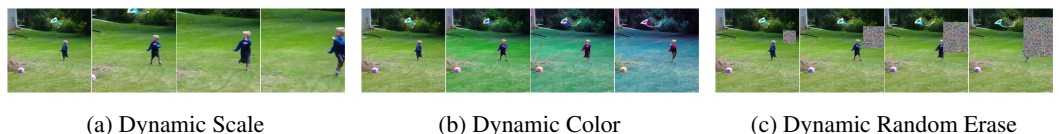

(a) Dynamic Scale          (b) Dynamic Color          (c) Dynamic Random Erase

Figure A5: Visualization of augmentation operations used for the extended search space of DynaAugment.

Table A3: Ablation study on additional operations in the extended search space. Experiments are conducted on Kinetics-100. Top-1 and Top-5 accuracies are described. RE: dynamic random erase, Scale: dynamic scale, Color: dynamic color.

| Baseline | Static | Dynamic | Dynamic, RE | Dynamic, Scale+RE | Dynamic, Scale+Color | Dynamic, Scale+Color+RE |
|---|---|---|---|---|---|---|
| RA | 72.2 / 90.0 | 73.4 / 90.2 | 73.3 / 90.4 | 73.6 / 90.5 | 73.3 / 90.8 | 73.6 / 90.8 |
| TA | 72.3 / 89.4 | 73.2 / 90.0 | 73.0 / 89.8 | 73.8 / 90.1 | 73.5 / 90.4 | 73.7 / 90.5 |

## A.4 DETAILS ON FOURIER SAMPLING

In this subsection, we describe the motivations and procedures of Fourier Sampling introduced in Section. 3.3. Like a mixture of augmentation described in AugMix (Hendrycks et al., 2019), Fourier Sampling is designed for the mixture of temporal variations. A basic unit for a temporal variation is a simple periodic signal. A periodic signal can be varied by modifying its frequencies.

Following Hendrycks et al. (2019), we first sample weights $w_b$ for the basis from the Dirichlet distribution so that the sum of the weight $f_b$ becomes 1. Next, for each basis, we randomly sample a frequency from the uniform distribution between 0.2 to 1.5. Fig. 3 (a) shows an example signal whose frequency is 0.2. As shown in the figure, because a signal with $f_b = 0.2$ is almost linear, the signal is selected as a lower frequency bound for Fourier Sampling. Fig. 3 (d) shows an example signal whose frequency is 1.5, and it is selected as a upper bound for Fourier Sampling by the heuristics. Because most of the video models take up to 32 frames, as described in Section. A.1., too high frequency can hurt temporal consistency. After sampling a frequency, a sinusoidal signal is generated, and after then, random offset $o_b$ and random amplitude $A$ are applied to make the signal more diverse into $x$ and $y$-axis. An amplitude is proportional to the static magnitude $M$ because more variations are required if a magnitude is large. Finally, the generated signals are integrated with Eq. (1). For the number of basis $C$, we experimentally confirm that there was no performance increase for values greater than 3 (See Section B.5.).

To achieve the best performance, manual search on the frequency, offset, amplitude, and number of basis is a possible strategy. However, it costs tremendous computational resources. Instead of giving it up, we chose a simple random sampling strategy, following search-free methods: TA (Müller & Hutter, 2021) or UA (LingChen et al., 2020).

## A.5 DETAILS ON THE SMOOTHNESS

In Table 6 (b), we compare DynaAugment's temporal variation (Fourier Sampling) with the three baselines: Linear, Sinusoidal, and Random. We visualize the three baselines in Fig. A6. In this figure, we set the $M = 5.0$ and use 32 frames for the visualization.

For Linear, we use the same policy for the amplitude and the offset with the Fourier Sampling (Fig. A6 (b)). For Sinusoidal, from the Fourier Sampling, we set the range of random frequency to scalar 1.0, and we use the number of basis $C = 1$ (Fig. A6 (d)). For Random, a magnitude for each frame is randomly sampled as in Fig. A6 (e).

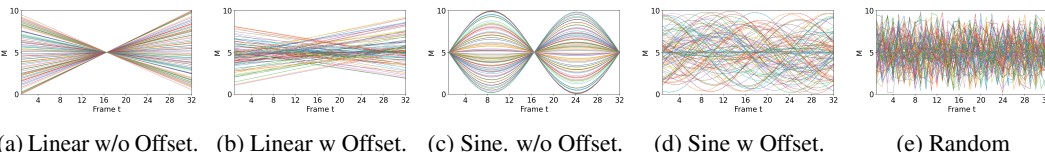

(a) Linear w/o Offset.  (b) Linear w Offset.  (c) Sine. w/o Offset.  (d) Sine w Offset.  (e) Random

Figure A6: Visualizations for Table 6.

### A.6  DETAILS ON THE AFFINITY AND THE DIVERSITY

In Table 6 (c), we use *the affinity* and *the diversity* as a evaluation metric for data augmentation methods. These metrics are introduced in Gontijo-Lopes et al. (2020). *The affinity* is calculated by the ratio between the validation accuracy of a model trained on clean data and tested on an augmented validation set (including stochastic operations), and the accuracy of the same model tested on clean data. *The diversity* is calculated by the the ratio of the final training loss of a model trained with a given augmentation (also, including randomness), relative to the final training loss of the model trained on clean data. The affinity is a metric for distribution shift and the diversity is a measure of augmentation complexity. (Gontijo-Lopes et al., 2020) find that performance is only improved when an augmentation operation increases the total number of unique training examples. The utility of these new training examples is informed by the augmentation's (high) affinity and (high) diversity.

### A.7  DETAILS ON THE CORRUPTED VIDEOS

Corrupted Kinetics validation set used in Fig. 4 is generated from the original validation set of Mini-Kinetics-200 (Xie et al., 2018) using following command: `ffmpeg -i [target_video] -c:v libx264 -preset slow -crf [QP]` . where ffmpeg library (Tomar, 2006) is used, `target_video` is a directory of original validation video, and `QP` is a quantization parameter of H.264 (Wiegand et al., 2003). RAFT (Teed & Deng, 2020) is used for optical flow visualization in Fig. 4 (b).

## B  ADDITIONAL RESULTS

### B.1  CONFIDENCES IN RESULTS

We run all the experiments three times unless specified, and all results in the manuscript are averaged over three runs. For all recognition results, the standard deviations of Top-1 accuracy are less than 0.19%. Most of cases, because DA's improvement gaps over static augmentations are greater than 0.19% except in some cases (*e.g.*, TDN-RA in K400), the superiority of DA is mostly guaranteed.

For transfer learning results, we describe standard deviations in Section B.4. We run all the experiments five times for the transfer learning tasks.

### B.2  MORE BASELINE RESULTS

In Table A4, we show the results of more baseline for RA (Cubuk et al., 2020), TA (Müller & Hutter, 2021), and UA (LingChen et al., 2020) in Kinetics-100. The results show that our baseline choice is reasonable because all these baselines (marked as underline and bold) show the best performances among the same augmentation methods. Especially, in RA, the performance reduces in the other hyper-parameters.

In Table A5, we compare DA results with more augmentation methods in Something-Something-v2 and UCF-101 datasets, such as, RMS (Kim et al., 2020b), MixUp (Zhang et al., 2018), VideoMix (Yun et al., 2020), and AutoAugment (AA) (Cubuk et al., 2019). For RMS, We use mean filter for the

Table A4: More Baseline Results on Kinetics-100: For, RA (N, M), N is the number of operations, and M is the magnitude. E means the extended search space. For RA, default probability is 1.0. Underline setting is the second best baseline, and **Bold** setting is the best baseline among the same augmentation method.

| Config. | Top-1/Top-5 | Config. | Top-1/Top-5 | Config. | Top-1/Top-5 |
|---|---|---|---|---|---|
| Baseline | 66.6 / 85.5 | | | | |
| RA (2, 9) | 72.2 / 90.0 | TA | 71.1 / 88.4 | UA | 70.3 / 88.1 |
| **RA-E (2, 9)** | 73.0 / 90.1 | TA-Wide | 72.3 / 89.4 | UA-Wide | 71.9 / 89.3 |
| RA-E (1, 9) | 71.8 / 89.1 | **TA-E** | 72.7 / 89.9 | **UA-E** | 72.8 / 89.7 |
| RA-E (2, 7) | 71.1 / 89.0 | | | | |
| RA-E (3, 9) | 71.7 / 90.0 | | | | |
| RA-E (4, 9) | 70.6 / 89.4 | | | | |
| RA-E (4, 7) $p = 0.5$ | 71.7 / 88.9 | | | | |
| RA (2, 9) Wide | 67.8 / 88.7 | | | | |
| RA (2, 9) + DA | 73.4 / 90.2 | TA + DA | 73.2 / 90.0 | UA + DA | 72.9 / 90.0 |
| **RA-E (2, 9) + DA** | 73.6 / 90.8 | **TA-E + DA** | 73.7 / 90.5 | **UA-E + DA** | 73.7 / 90.3 |

pooling operation, and random sampling distribution is Gaussian whose mean is $1.0$ and standard deviation is $0.5$. RMS operation is inserted before the 3rd batch normalization layer of all residual blocks. The mixing ratio between two data samples is sampled from `Beta`$(1.0, 1.0)$ that is identical to `Uniform`$(0.0, 1.0)$. The mixing is occurred within the mini-batch. For VideoMix, the box coordinates are uniformly sampled following the original implementation. For video, MixUp and VideoMix are applied identically across frames. In the case of CutMix (Yun et al., 2019), it is identical to the spatial version of VideoMix. For AA, we use searched policy from the ImageNet as in RA.

Table A5: Comparison with more baselines: RMS (Kim et al., 2020b), MixUp (Zhang et al., 2018), VideoMix (Yun et al., 2020), and AutoAugment (AA) (Cubuk et al., 2019), on Something-Something-v2, UCF-101, and Diving-48 datasets. We report Top-1 and Top-5 accuracies.

| Dataset | Something-v2 | | UCF-101 | Diving-48 |
|---|---|---|---|---|
| Model | SlowFast8×8 | SlowFast16×8 | S3D-G | S3D-G |
| Baseline | 61.4 / 85.8 | 63.0 / 88.5 | 72.0 / 90.6 | 62.8 / 93.6 |
| RA | 63.1 / 87.3 | 64.4 / 88.7 | 82.8 / 96.4 | 63.6 / 93.8 |
| RA+DA | **65.0 / 89.0** | 65.5 / 89.5 | **85.3 / 97.2** | 70.5 / 95.6 |
| TA | 63.5 / 89.0 | 64.7 / 88.6 | 81.6 / 96.3 | 68.3 / 95.1 |
| TA+DA | 64.8 / 88.9 | **65.8 / 89.8** | 82.8 / 95.7 | **72.5 / 95.7** |
| UA | 63.1 / 88.7 | 64.6 / 88.5 | 82.5 / 96.1 | 67.8 / 95.0 |
| UA+DA | 64.0 / 88.9 | 65.7 / 89.6 | 82.9 / 95.6 | 70.0 / 94.9 |
| RMS (Kim et al., 2020b) | 62.1 / 86.8 | 63.9 / 88.0 | - | - |
| MixUp (Zhang et al., 2018) | 62.4 / 87.3 | 63.5 / 88.4 | 76.7 / 91.8 | **65.6 / 93.8** |
| VideoMix (Yun et al., 2020) | **63.2 / 88.2** | 64.2 / 88.4 | 73.8 / 91.9 | 63.0 / 93.4 |
| AA (Cubuk et al., 2019) | 63.1 / 87.4 | **64.6 / 88.6** | **81.7 / 94.9** | 63.1 / 93.6 |

## B.3 UCF/HMDB RESULTS ON THE PRE-TRAINED MODELS

We describe the results on UCF-101 (Soomro et al., 2012) and HMDB-51 (Kuehne et al., 2011) datasets using ImageNet or Kinetics pre-training in Table A6. From the results, we can check the effects of augmentations in terms of model pre-training. Although the improvement of performance are reduced compare to *training-from-scratch* through the pre-training, DA still increases the performances.

Table A6: Results on UCF-101 and HMDB-51 datasets with different pre-training settings.

| Dataset | Model | Baseline | RA | RA + DA | TA | TA + DA | UA | UA + DA |
|---------|-------|----------|-----|---------|-----|---------|-----|---------|
| UCF-101 | S3D-G from Scratch | 72.0 / 90.6 | 82.8 / 96.4 | **85.3 / 97.2** | 81.6 / 96.3 | **82.8 / 95.7** | 82.5 / 96.1 | **82.9 / 95.6** |
| | I3D-16×8 from ImageNet | 79.7 / 93.7 | 87.4 / 97.8 | **88.7 / 98.0** | 86.2 / 97.2 | **87.3 / 97.5** | 85.8 / 97.3 | **87.6 / 97.9** |
| | I3D-16×8 from Kinetics | 92.4 / 98.9 | 93.5 / 98.7 | **94.5 / 98.9** | 94.4 / 99.0 | **94.5 / 99.1** | 94.0 / 99.2 | **94.8 / 99.3** |
| HMDB-51 | S3D-G from Scratch | 40.1 / 72.6 | 48.5 / 78.4 | **54.8 / 83.7** | 47.7 / 78.9 | **51.9 / 80.7** | 49.9 / 79.8 | **52.4 / 78.9** |
| | I3D-16×8 from ImageNet | 44.9 / 72.3 | 53.6 / 80.9 | **54.4 / 83.8** | 51.6 / 79.7 | **52.7 / 81.4** | 50.9 / 79.6 | **51.2 / 79.9** |
| | I3D-16×8 from Kinetics | 62.9 / 86.0 | 64.9 / 88.5 | **66.8 / 88.9** | 64.7 / 87.5 | **67.0 / 89.0** | 67.0 / 89.9 | **67.5 / 89.9** |

Table A7: Full results with standard deviations of Table 4.

(a) Action segmentation results on Breakfast dataset.

| | | F1 | | |
|---------------|------|---------|---------|---------|
| Configuration | Acc. | @0.10 | @0.25 | @0.50 |
| SO-50 | 59.0 | 54.7$_{\pm2.0}$ | 49.2$_{\pm1.7}$ | 37.6$_{\pm1.8}$ |
| SO-50+TA | 62.0 | 58.8$_{\pm1.2}$ | 52.7$_{\pm1.4}$ | 40.2$_{\pm1.6}$ |
| SO-50+TA+**DA** | **64.3** | **60.1**$_{\pm1.2}$ | **54.9**$_{\pm1.6}$ | **42.8**$_{\pm1.5}$ |
| SO-50+UA | 62.3 | 59.1$_{\pm1.1}$ | 53.7$_{\pm1.6}$ | 40.8$_{\pm1.4}$ |
| SO-50+UA+**DA** | **64.7** | **60.6**$_{\pm1.0}$ | **55.5**$_{\pm1.4}$ | **43.2**$_{\pm1.5}$ |

(b) Object detection results on MOT17det dataset.

| Configuration | AP | AP50 | AP75 | APs |
|---------------|-----|------|------|-----|
| Swin-T | 30.3$_{\pm0.9}$ | 63.7$_{\pm1.3}$ | 25.1$_{\pm1.4}$ | 4.1$_{\pm0.8}$ |
| Swin-T+TA | 29.1$_{\pm1.3}$ | 62.1$_{\pm1.6}$ | 23.2$_{\pm2.3}$ | 3.7$_{\pm0.7}$ |
| Swin-T+TA+**DA** | **31.3**$_{\pm0.6}$ | **65.3**$_{\pm0.8}$ | **26.3**$_{\pm1.0}$ | **5.7**$_{\pm0.3}$ |
| Swin-T+UA | 29.4$_{\pm1.1}$ | 62.9$_{\pm1.5}$ | 23.4$_{\pm1.6}$ | 4.1$_{\pm1.5}$ |
| Swin-T+UA+**DA** | **31.4**$_{\pm0.7}$ | **65.5**$_{\pm0.5}$ | **25.9**$_{\pm1.8}$ | **4.4**$_{\pm0.1}$ |

## B.4 MORE RESULTS ON TRANSFER LEARNING

In Table A7, we describe TA results for Breakfast (Kuehne et al., 2014) dataset and UA results for MOT17det (Milan et al., 2016) dataset with their standard deviations. These results show the similar trend as in the main manuscript.

## B.5 MORE RESULTS ON FOURIER SAMPLING

In Table A8, we describe more ablation studies on Fourier Sampling as explained in Section A. 4. in UCF-101 dataset. For the number of basis, any choice shows superiority over the baseline and the static version. For the amplitude, a wide range shows better and more stable results. For the range of frequencies, because the other setting also shows similar performances, we choose the range by the heuristic. Adding random offset can generate more diverse signals, so the performance is increased. Gaussian smoothing over random variations can be a substitute for Fourier Sampling. However, it is not only inexplicable but also difficult to improve performance when implemented simply.

Table A8: More ablation studies on Fourier Sampling. We describe Top-1 and Top-5 accuracies on UCF-101 dataset. **Bold** indicates our default setting.

| Config. | Acc. | Config. | Acc. |
|---------|------|---------|------|
| Baseline | 72.0 / 90.6 | | |
| $C = 1$ | 84.4 / 97.1 | | |
| $C = 2$ | 84.7 / 97.0 | $f_b$=**[0.2 ∼ 1.5]** | **85.3 / 97.2** |
| $C = 3$ | **85.3 / 97.2** | $f_b$=[0.2 ∼ 1.0] | 85.1 / 97.0 |
| $C = 4$ | 85.2 / 97.1 | $f_b$=[0.5 ∼ 2.0] | 84.6 / 96.9 |
| $A$=[0.0 ∼ 0.5] | 83.5 / 96.8 | **w Offset** | **85.3 / 97.2** |
| $A$=[0.5 ∼ 1.0] | 83.9 / 97.0 | w/o Offset | 84.7 / 96.5 |
| $A$=**[0.0 ∼ 1.0]** | **85.3 / 97.2** | Random+Gaussian | 83.1 / 96.4 |

Exploring the additional hyperparmeters of DA may show better performance. However, its optimum values may differ over models and datasets, which results in extensive computational search costs. Instead, in this paper, our main objective is to show that there is room for further performance improvements by simply making the augmentation operations dynamic over static, rather than to reach the state-of-the-art numbers per-model and per-dataset.

## B.6 RESULTS ON DIFFERENT FRAME RATES

To verify the effects of DA on different frame rates, we test on Kinetics-100 for three models that have different frame rates: SlowOnly-8x8, 8x4, and 8x2. For each model, we apply three frequency ranges: (0.2, 1.0), (0.5, 1.5), and (1.0, 2.0), where (a, b) indicates that is randomly sampled from the minimum (a) to maximum (b) of $f_b$ in Eq. (1). TA is used for baseline augmentation. The results in Table A9 show that the model that takes the frame faster (SlowOnly-8x8) performs better in higher frequencies and vice versa.

Table A9: Top-1 accuracy on Kinetics-100. Note that (0.2, 1.5) is a default setting of DA.

| Model | DA (0.2, 1.0) | DA (0.5, 1.5) | DA (1.0, 2.0) |
|---|---|---|---|
| SlowOnly-8x8 | 73.5 | **73.7** | 73.5 |
| SlowOnly-8x4 | **72.8** | 72.6 | 72.5 |
| SlowOnly-8x2 | **71.7** | 71.4 | 71.0 |

## B.7 RESULTS ON DIFFERENT DATASET CHARACTERISTICS

To check that a more dynamic dataset (*e.g.* Diving-48) benefit from a larger augmentation strength than a less dynamic one (*e.g.* UCF101), we attach the results in Table A10 of the experiment to both types of datasets. UCF-101 is the less dynamic dataset, and Diving-48 is the more dynamic dataset. We used default hyperparameters of Fourier Sampling but changed the frequency sampling ranges. In the table, (a, b) indicates that is randomly sampled from the minimum (a) to maximum (b). RA is used for baseline augmentation.

As in the table, results on dynamic dataset show further improvements with higher frequency sampling range, *e.g.*, DA (1.0, 2.0), which empirically proves our intuitions. But because the differences are not dramatic, we chose simplicity (same parameters for all experiments) as the default.

Table A10: Top-1 accuracy on UCF-101 and Diving-48. Note that (0.2, 1.5) is a default setting of DA.

| Dataset | RA | DA (0,2, 1.0) | DA (0,2, 1.5) | DA (0,5, 1.5) | DA (1.0, 2.0) |
|---|---|---|---|---|---|
| UCF-101 | 82.8 | 85.1 | **85.3** | 84.6 | 84.8 |
| Diving-48 | 63.6 | 68.8 | 70.5 | 70.7 | **70.9** |

## B.8 QUALITATIVE RESULTS

In Fig. A7, we show additional qualitative results in the case of extreme temporal variations. We order four examples from top to bottom, and from left to right. In the first example (top left), the confidence of correct class (*Wind Surfing*) is hugely increased in DA compared to RA. In the second example (top right), compared to RA, Top-5 predictions are changed into relevant classes (for example, *feeding goats*, *riding elephant*, or *walking the dog*). In the third example (bottom left), the class ambiguity is resolved, confidence of *playing volleyball* is decreased compared to that of RA, and another relevant classes are predicted, such as *roller skating* and *ice skating* due to the floor reflection. In the last example (bottom right), scene bias (*e.g. crawling baby*) is removed from the results of RA, and another relevant class is predicted (*e.g. head banging*).

## B.9 DYNAAUGMENT FOR SELF-SUPERVISED LEARNING

As mentioned in the conclusion and limitation section, our study is limited to supervised learning. However, we conduct a simple initial experiment to apply DynaAugment in other settings (*e.g.* self-supervised learning based on data augmentations). In the image domain, strong data augmentation (*e.g.* RA) has been used instead of basic data augmentations (*e.g.*, Crop, Color, Flip, or Blur) in some works (Tian et al., 2020; Bai et al., 2021), and they have shown promising results. Based on these

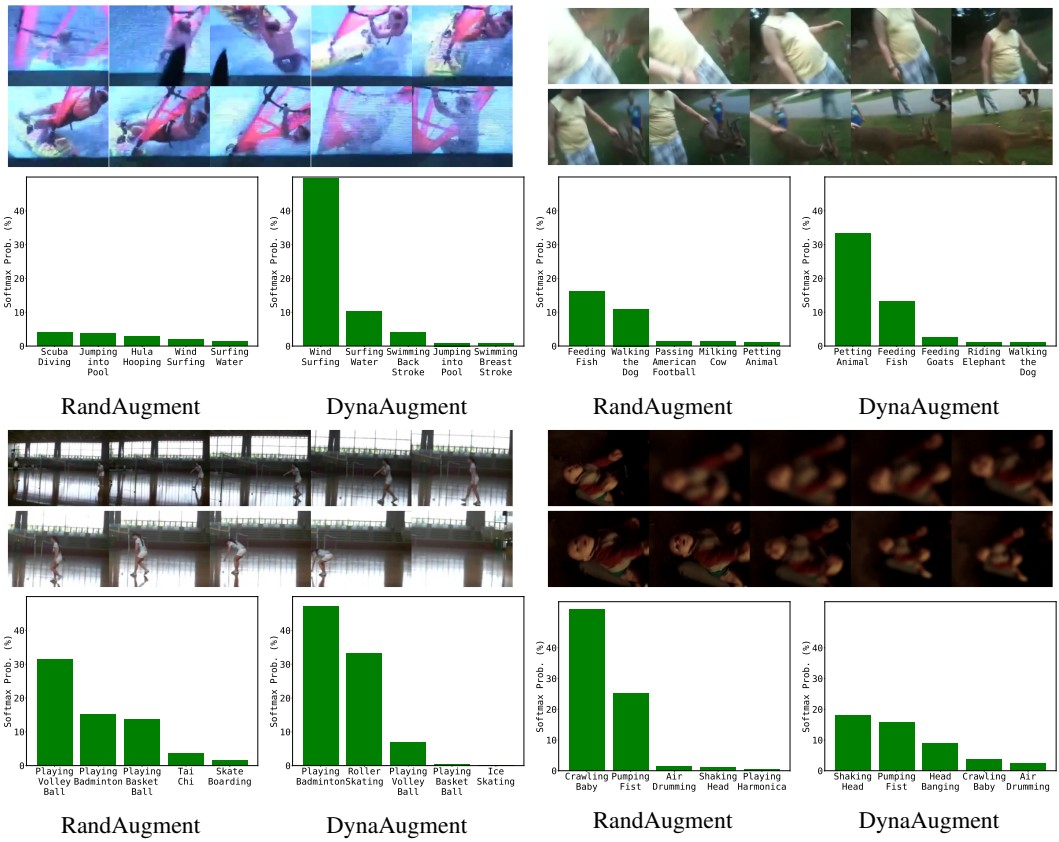

Figure A7: Qualitative Results. From left to right, top to bottom, the ground truth classes are *Wind Surfing*, *Petting Animal*, *Playing Badminton*, and *Shaking Head*.

observations and settings, we experiment with BYOL (Grill et al., 2020), the representative data augmentation-based SSL method for video domains.

We use SlowOnly-50 as the backbone, and BYOL is pre-trained on Kinetics-100 using RA and RA+DA, and they are fine-tuned on UCF-101 to measure their performances (Top-1 and Top-5). For RA and RA+DA, RA or RA+DA is used as default augmentation instead of BYOL's default augmentations.

The result in Table A11 proves clear effectiveness of DA in self-supervised learning.

Table A11: Results on UCF-101 from the differntly learned BYOL models (on Kinetics-100 pre-training).

|  | Original Aug. | RA | DA |
|---|---|---|---|
| Top-1/5 | 76.05 / 93.87 | 79.75 / 95.59 | **80.78 / 95.76** |

