# OpenReview forum: "Exploring Temporally Dynamic Data Augmentation for Video Recognition"
_ICLR.cc/2023/Conference — ICLR 2023 notable top 25%_

### Official Review · Reviewer_Ntrm · 2022-10-24

**Confidence:** 4
**Correctness:** 3
**Technical Novelty And Significance:** 3
**Empirical Novelty And Significance:** 3
**Recommendation:** 8

**Clarity, Quality, Novelty And Reproducibility:**

This paper is well organized and the technical novelty is satisfactory.
Since it is implemented upon RA, UA, and TA, it should be easily reproduced.

**Strength And Weaknesses:**

Strength:
-The motivation is straightforward and well-presented. The solution is simple yet effective and it can be seamlessly applied to the existing auto-augmentation methods such as RA, UA, and TA.
-The experiments are enough to prove the effectiveness of the proposed Fourier sampling for dynamic augmentation. Extensive ablation studies are provided in the appendix.

Weaknesses:
-The baselines on experiments are somewhat weak. For example, Swin-Tiny and Uniformer-S in Table1 and SlowOnly in Table 3/4 have relatively low performance at the beginning. Stronger baselines are expected.



**Summary Of The Paper:**

This paper explores automatic data augmentation by introducing a simple yet effective Fourier sampling. It provides dynamic, smooth, and diverse augmentation, especially for videos with a time dimension. Many experiments on large-scale, small-scale, and fine-grained video action tasks are conducted and it shows an almost consistent improvement over multiple benchmarks.

**Summary Of The Review:**

Considering the extensive experiments the authors provide, the effectiveness of the proposed sampling though simple is well proven. I feel positive for this paper at this moment.

After reading the rebuttal, I feel that my concerns have been well addressed. More strong baselines have been included to make this paper more solid.

---

> ### Author Response · Authors · 2022-11-16
> **Authors Response to Reviewer Ntrm**
>
> Thank you for commenting on the positive side of our paper, especially on the presentation and experimental aspects.
>
> [About the Stronger Baselines]
>
> We agree with the reviewer's comment that experiments with stronger baselines will make our paper more solid and comprehensive. Therefore, we updated experiments with stronger baselines, e.g., Uniformer-Base on Kinetics-400 in Table 1 (Note that Swin-Base and Uniformer-Base were already used in Something-v2 Dataset in Table 1). We also updated experiments for temporal action localization on the THUMOS dataset in Table 3, e.g., SlowFast-50.
>
> The updated results also demonstrated that our method is generalizable across model types, scales, and tasks.
>
> The rest of the results (e.g. Swin-B on K400 / Stronger Backbones for Breakfast) will be updated before the rebuttal deadline and will be notified to the reviewer, or will be updated in the final copy if they are not completed before then (Note that training those models requires more than 4~5 days).

---

> > ### Author Response · Authors · 2022-11-18
> > **Authors Respones to Reviewer Ntrm (New Results are Updated)**
> >
> > Dear Reviewer,
> >
> > We updated the results for Swin-Base on Kinetics-400 in the last row of Table 1: Weak-Aug. Baseline.
> >
> > The results showed that our method is still effective on the larger models.
> >
> > In the final copy, we will update the remaining experiments (stronger baseline for Table 4).

---

### Official Review · Reviewer_RUAa · 2022-10-24

**Confidence:** 5
**Correctness:** 4
**Technical Novelty And Significance:** 2
**Empirical Novelty And Significance:** 3
**Recommendation:** 6

**Clarity, Quality, Novelty And Reproducibility:**

The clarity and the quality of the paper presentation are on a high level. It has nice visualizations with an easy-to-follow structure. The paper and its supplementary materials contain a detailed description of the models and experiments. So I would say that the work and the results are reproducible. However, the paper has a little bit of limited technical novelty.

**Strength And Weaknesses:**

**Strengths**:

1) The paper is very well-written with good structure and nice visualizations. The motivation behind DynaAugment is clear and fitted to the nature of the video domain.
2) The proposed method is based on a strong theoretical framework (Fourier Sampling) with the application for general dynamic data augmentation with smooth and diverse properties.
3) The ablation study shows the benefits of the chosen methodology for dynamic variations over other possible alternatives.
On the large and diverse set of video datasets DynaAugment outperforms image-based data augmentation techniques using even different backbones including CNNs and transformers. The most significant improvement is shown on small-scale and fine-grained datasets of the most practical interest.
4) DynaAugment is also useful for transfer learning other than video action recognition tasks such as action localization, action segmentation, video object detection, and self-supervised learning
5) DynaAugment helps in the case of corrupted videos showing better results than other data augmentations.

**Weaknesses**:
1) A little bit of limited technical novelty which consists mainly of temporally-smoothed parameterized data augmentation.


**Summary Of The Paper:**

The paper tackles the problem of data augmentation for training video models. It argues that existing augmentation techniques just mostly extend image augmentation strategies and don't fully utilize the temporal dynamics of videos. The paper proposes a new data augmentation framework specifically designed for videos (DynaAugment) that changes the magnitude of augmentation operations for each frame with the help of Fourier Sampling. The experiments are conducted on a diverse set of video datasets including large-scale, small-scale, and fine-grained. DynaAugment improves results in every scenario in combination with existing image-based data augmentation techniques.

**Summary Of The Review:**

I have a favorable opinion of the paper and I think the paper deserves the acceptance rating.

---

> ### Author Response · Authors · 2022-11-16
> **Authors Response to Reviewer RUAa**
>
> Thank you for commenting on the positive side of our paper, especially on the writing and experimental aspects.
>
> [About the Technical Novelty]
>
> We can summarize the technical novelty of our paper as follows:
>
> - Our method can generate temporally smooth, diverse, and realistic temporal variations. In addition to the smooth and diverse characteristics mentioned by the reviewer, we want to emphasize the “realistic” characteristics. And this is the most important motivation for designing Fourier Sampling, over other naive smooth variations (in Table 5(c) and Random+Gaussian in Table A8.).
>
> - The expanded search space (described on Page 5, Sec 4.5, and Table 5a) is also our new module for video-level data augmentation. Motivated from real-world videos, our new operations were added for the extended search space. Appendix A.3 contains more details.
>
> We apologize for not clearly emphasizing these things in the paper and thank you for pointing this out. We believe that introducing an entirely new method is not the only way to be a technically novel work. Instead, we believe proposing a simple way to solve an essential but neglected issue of existing methods, i.e., temporal dynamics, is also novel.

---

### Official Review · Reviewer_LMkg · 2022-10-25

**Confidence:** 4
**Correctness:** 4
**Technical Novelty And Significance:** 4
**Empirical Novelty And Significance:** 4
**Recommendation:** 8

**Clarity, Quality, Novelty And Reproducibility:**

[Clarity] The clarity of the manuscript is good.
[Quality] The overall quality of this paper is impressive.
[Novelty] The novelty of this submission has reached the acceptance standard of ICLR, in my opinion.
[Reproducibility] It is hard to judge the reproducibility of the submission since no code is provided. Based on the implementation details, the re-implementation of the method should not be that hard.


**Strength And Weaknesses:**

Strengths:

1. The work is fundamental and meaningful to the video research community, since most of video downstream tasks involve data augmentation strategies.
2. Good insight and meaningful motivation. And clear illustration of the main idea.
3. Extensive and consistent experimental support, including most of the commonly-used benchmark datasets. Remarkably, some of the experiments are interesting and novel. E.g., classification on corrupted videos.
4. Clear writing style and good presentation.


**Summary Of The Paper:**

This paper presents a simple yet effective video data augmentation framework, called DynaAugment, to improve the robustness and accuracy of several video recognition tasks, e.g., action classification, action segmentation, action localization, video object detection. The proposed data augmentation framework respects and simulates the real-world video's temporal variations, instead of naively extend the image augmentation methods, and thus it enjoys good diversity and temporal smoothness. Specifically, a novel sampling function called Fourier Sampling is presented to simultaneously achieve the two goals, i.e., diversity and smoothness. With extensive ablation studies and experiments on multiple downstream tasks such as action recognition and corrupted video classification, impressive superiority of the proposed method is demonstrated.

**Summary Of The Review:**

The overall quality of this submission is good. Please see "Strengths and Weaknesses" above.
I tend to accept this paper.

---

> ### Author Response · Authors · 2022-11-16
> **Authors Response to Reviewer LMkg**
>
> We are very grateful for the reviewer's positive comments.
>
> [About the Reproducibility]
>
> As the reviewer pointed out, our method is not difficult to implement based on the existing augmentation algorithms. Nevertheless, we will release the code as open source to reproduce the performances. Until then, we revised the paper with all possible detailed implementation information in Section A.1.
>
> Specifically, we implemented our augmentation module on top of the public codebases [A-E] for video action recognition. The modifications we made were 1) augmentation part and 2) hardware-specific part and configurations (e.g. GPUs and Batch Size). For downstreams, we also used official implementations [F-G] based on our extracted features.
>
> [A] https://github.com/facebookresearch/SlowFast
>
> [B] https://github.com/mit-han-lab/temporal-shift-module
>
> [C] https://github.com/MCG-NJU/TDN
>
> [D] https://github.com/SwinTransformer/Video-Swin-Transformer
>
> [E] https://github.com/Sense-X/UniFormer/tree/main/video_classification
>
> [F] https://github.com/frostinassiky/gtad
>
> [G] https://github.com/yabufarha/ms-tcn

---

### Official Review · Reviewer_wBSm · 2022-10-25

**Confidence:** 4
**Clarity, Quality, Novelty And Reproducibility:** Simple, clear, novel idea. Open sourc…
**Correctness:** 3
**Technical Novelty And Significance:** 3
**Empirical Novelty And Significance:** 3
**Recommendation:** 8

**Strength And Weaknesses:**

**Strengths**
* Well-written paper and well-motivated method.
* Convincing extensive experiments on a wide range of video recognition tasks.
* Simple widely applicable method.

**Weaknesses**
* Open source implementations of the proposed augmentations *in a range of existing frameworks* (e.g. numpy, tensorflow, etc) would greatly help adoption of the proposed methods.

**Summary Of The Paper:**

The paper proposes a generalisation of commonly used image augmentation to video, paying specific attention to temporal consistency and smoothness of the said augmentations. Efficacy of the proposed generalisation is supported by strong empirical results on a wide range of video understanding tasks.

**Summary Of The Review:**

Clear simple video augmentation technique that is demonstrated to consistently improve model performance in upstream and downstream tasks.

---

> ### Author Response · Authors · 2022-11-16
> **Authors Response to Reviewer wBSm**
>
> We are very grateful for the reviewer's positive comments.
>
> [Open Source Implementations / Implementation Details]
>
> As the reviewer pointed out, we will release the code as open source to reproduce the performances. Until then, we revised the paper with all possible detailed implementation information in Section A.1.
>
> Specifically, we used PyTorch, and we implemented our augmentation module on top of the public codebases [A-E] for video action recognition. The modifications we made were 1) augmentation part and 2) hardware-specific part and configurations (e.g. GPUs and Batch Size). For downstreams, we also used official implementations [F-G] based on our extracted features.
>
> [A] https://github.com/facebookresearch/SlowFast
>
> [B] https://github.com/mit-han-lab/temporal-shift-module
>
> [C] https://github.com/MCG-NJU/TDN
>
> [D] https://github.com/SwinTransformer/Video-Swin-Transformer
>
> [E] https://github.com/Sense-X/UniFormer/tree/main/video_classification
>
> [F] https://github.com/frostinassiky/gtad
>
> [G] https://github.com/yabufarha/ms-tcn

---

### Decision · Program_Chairs · 2023-01-20

**Decision:**

Accept: notable-top-25%

**Justification For Why Not Higher Score:**

- Limited scope: Impact limited to video understanding community dealing with primarily action recognition tasks
- No open-source implementation in PyTorch/JaX/TF

**Justification For Why Not Lower Score:**

- Solid publication, practical applications in video understanding, very likely to be adapted by the community.

**Metareview: Summary, Strengths And Weaknesses:**

The authors propose a data augmentation strategy for training video understanding models which focus on temporal consistency, diversity, and smoothness. The magnitude of augmentation operations on each frame is modulated by Fourier Sampling. The proposed method shows consistent improvement across many classic benchmarks. The reviewers were appreciated the significance, clarity, and novelty of the proposed method, as well as the consistent improvements in practice. I will recommend the acceptance of this work.

**Note From Pc:**

if the above contains the word "oral" or "spotlight" please see: "oral" presentation means -> notable-top-5% and "spotlight" means -> notable-top-25%. As stated in our emails, we are disassociating presentation type from AC recommendations